# An Adaptive Remote Sensing Image-Matching Network Based on Cross Attention and Deformable Convolution

Peiyan Chen †, Ying Fu †, Jinrong Hu, Bing He *, Xi Wu and Jiliu Zhou

School of Computer Science, Chengdu University of Information Technology, Chengdu 610225, China;
chenpeiy2023@163.com (P.C.); fuying@cuit.edu.cn (Y.F.); hjr@cuit.edu.cn (J.H.); wuxi@cuit.edu.cn (X.W.);
zhoujl@cuit.edu.cn (J.Z.)
* Correspondence: bingh@cuit.edu.cn
† These authors contributed equally to this work.

**Abstract:** There are significant background changes and complex spatial correspondences between multi-modal remote sensing images, and it is difficult for existing methods to extract common features between images effectively, leading to poor matching results. In order to improve the matching effect, features with high robustness are extracted; this paper proposes a multi-temporal remote sensing matching algorithm CMRM (CNN multi-modal remote sensing matching) based on deformable convolution and cross-attention. First, based on the VGG16 backbone network, Deformable VGG16 (DeVgg) is constructed by introducing deformable convolutions to adapt to significant geometric distortions in remote sensing images of different shapes and scales; second, the features extracted from DeVgg are input to the cross-attention module to better capture the spatial correspondence of images with background changes; and finally, the key points and corresponding descriptors are extracted from the output feature map. In the feature matching stage, in order to solve the problem of poor matching quality of feature points, BFMatcher is used for rough registration, and then the RANSAC algorithm with adaptive threshold is used for constraint. The proposed algorithm in this paper performs well on the public dataset HPatches, with MMA values of 0.672, 0.710, and 0.785 when the threshold is selected as 3–5. The results show that compared to existing methods, our method improves the matching accuracy of multi-modal remote sensing images.

**Keywords:** multi-modal remote sensing images; image registration; cross-attention; deformable convolution; homography matrix





## 1. Introduction

In recent years, the progress of remote sensing technology has enabled people to use various sensors to acquire rich remote sensing images. Different satellite sensors can provide multi-modal remote sensing images with multi-temporal, multi-resolution, and multi-spectrum properties for the same area. These multi-modal remote sensing data have good complementarity and make up for the deficiency of a single data source. Remote sensing image registration can establish the corresponding relationship between two or more remote sensing images obtained, and is widely used in monitoring environmental change, planning urban areas, and testing land cover. However, due to complex background changes caused by time, weather, and natural disasters, objects may undergo scale changes and deformation, which makes multi-modal registration challenging. The solution to this problem is of great practical significance because it determines whether the subsequent change detection and image fusion are successful.

Traditional image registration methods can be divided into two categories: (1) Feature-based methods [1] and (2) region-based methods [2]. The feature-based method is generally divided into three separate stages: feature detection, feature description, and feature matching. In the feature detection stage, significant points such as corner points should be detected as interest points, and then local descriptors should be extracted around these

corner points. Through the nearest neighbor search [3] or a more complex matching algorithm, the corresponding relationship between the two can be found to complete the registration. Scale Invariant Feature Transform (SIFT) [4] is a popular and widely used feature-based registration method, but the matching time is long, and it is easy to cause detail loss. Based on the advantages of SIFT, many improved versions have been proposed to enhance the performance of feature extraction, descriptor, and matching, such as RootSIFT [5], PCA-SIFT [6], ASIFT [7], etc. Common point feature extraction methods include Harris [8] operator, SURF [9], ORB [10], and FAST [11]. However, the hand-designed descriptors cannot accurately deal with the influence of non-optical linear illumination, shadow, and block, which are not good for the matching of remote sensing images with complex changes.

Region-based image registration is also known as template matching, that is, to find the most similar part of the input image and the template image, and its feature extraction and feature matching are carried out synchronously. However, this method is very sensitive to the difference in gray information between images. The normalized correlation coefficient has linear invariance to gray change and has been widely used in remote sensing image registration [12,13]. Some scholars also use structural features to construct similarity measurements for matching. Ye et al. used structural features to construct the Histogram of Orientated Phase Congruency (HOPC) [14], which was successfully applied to multi-modal image matching. Recently, they proposed CFOG [15] as an extension of HOPC.

The matching methods described above all require the artificial design of feature descriptors, and the information is redundant. With the increasing maturity of deep learning, the method of extracting key points and making feature descriptors by neural network has been widely used [16–18]. By learning semantic features in sample labels, the extracted features can be more robust. SCNN [19] designed a convolutional network combining a twin network with an S-Harris corner detector to improve the image-matching performance of remote sensing images with complex background changes. Compared with the traditional detect-then-describe method (extracting key points before making descriptors), D2-net [20] designed a description-and-detect strategy (extracting key points and descriptors at the same time), which made the feature module and description block highly coupled. This method performed well in image matching with day–night changes and large angle changes. SuperGlue [21] combined feature detectors and matcher as a complete pipeline, combining detection and matching into a network to improve matching accuracy. Patch2Pix [22] proposed a new angle to learn correspondence. This proposal directly optimized features from the matching network without clearly defining feature points. However, due to its patch-dependent principle, it may lead to the loss of global context information. LoFTR [23] borrowed from Transformer [24] and used the self-attention layer and mutual information layer [25] to obtain feature descriptors of two images, which can produce high-quality matching results in areas with low texture, motion blur, or repetitive image patterns. In response to the deformable images, Dai [26] increased an offset in CNN convolution kernels and built a deformable convolution network (deformable convolutional network, DCN).

After the rough feature matching is completed, the common RANSAC [27] algorithm eliminates the error points in registration. However, the selection of interior points depends on self-set parameter thresholds, and different types of images are different, which makes the setting of thresholds particularly important. MSAC [28], a modified version of RANSAC, uses the median instead of the mean as the cull threshold, enabling the algorithm to automatically configure corresponding parameters based on the characteristics of the data. MLESAC [29] is also modified on the basis of RANSAC, which uses maximum likelihood [30] to estimate the parameters of the model and provides a more robust and accurate solution than RANSAC, especially when there is a lot of noise in the data processed. In order to better solve this problem, an adaptive threshold algorithm is proposed in this paper, which can better screen out high-quality matching pairs.

The work of this paper can be summarized into the following three points:

1.　For multi-modal remote sensing image registration, this paper proposes an end-to-end trainable convolutional neural network CMRM, which can simultaneously detect features and extract dense feature descriptors;
2.　In order to better capture the relationship between global features and local features, a cross-attention mechanism is introduced; In order to extract effective features of deformable remote sensing images, deformable convolution blocks are added. The focus will be on learning how to extract effective feature points from remote sensing images with complex background changes;
3.　This paper designs an adaptive threshold RANSAC purification algorithm, which can automatically adjust threshold values according to data characteristics so as to screen out high-quality matching pairs and improve registration accuracy.

Through the above improvements, the method proposed in this paper can effectively extract significant feature points and adaptively eliminate mismatching points. Compared with existing popular algorithms, the algorithm in this paper not only improves the registration accuracy, but also improves the real-time registration. The rest of this paper is as follows: In Section 2, the design process of the proposed algorithm is described in detail. In Section 3, the results are given and analyzed in detail. Section 4 summarizes this paper.

## 2. Methodology

Effectively improving the matching performance of multi-temporal remote sensing images depends on reducing the influence of complex background changes on images, so this paper will carry out experimental design from the following four aspects: (1) VGG16 [31] is selected as the backbone network and deformable convolution layers and cross-attention are introduced for feature extraction; (2) the network is trained by using the existing remote sensing image dataset so that the feature extraction network can learn the robust expression of features; (3) building a qualified matcher is the key to the registration, and in this paper, the brute force matching algorithm (BFMatcher) will be chosen to complete the initial matching; and (4) after the initial matching, the error points will be eliminated by the adaptive threshold RANSAC algorithm. The above image registration workflow is fully automated. The network structure diagram is shown in Figure 1.

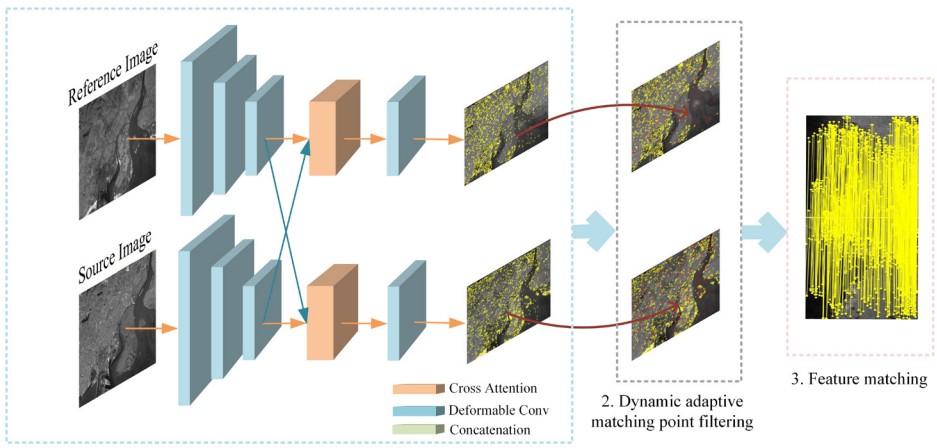

1. Feature points detection
Feeding the outputs of deformable convolutional lays into the cross attention, the output of the cross attention is uesd to compute attention weights between the two input feature maps

**Figure 1.** The proposed model structure of this paper.

### 2.1. Feature Extraction Network

Feature detection and feature descriptor generation are crucial steps for pixel-level registration of multi-modal remote sensing images. In this study, to better extract features and generate feature descriptors, we adjusted the VGG16 network structure for multi-modal remote sensing image matching and developed a deformable VGG16 model called

DeVgg (Deformable VGG16). To balance strong feature extraction capability with precise localization, we modified the backbone through the following steps.

First, four deformable convolution layers were constructed, which were placed behind the first, second, third, and fourth layers of VGG16 (*Pool1*, *Pool2*, *Pool3*, and *Pool4*). The deformable convolution layer consists of two parts. In the upper part, offset fields were introduced on the input feature maps to enable flexible sampling of the convolution kernels in the vicinity of the current position. In the lower part, deformable convolution was utilized to generate the output feature maps. The resulting feature maps have the same dimensions as the input feature maps. The deformable convolution formula is as shown in Formula (1):

$$y(P_0) = \sum_{p_n \in R} w(P_n) x(P_0 + P_n + \Delta P_n) \tag{1}$$

where $\Delta p_n (n = 1, 2, \ldots, N)$ is the offset of point $p_n$. Compared with traditional convolution, the shape and size of DeVgg's kernel can be dynamically changed according to input, which can provide more flexible and powerful capabilities for feature extraction in remote sensing images. The comparison between deformable convolution and traditional convolution is shown in Figure 2.

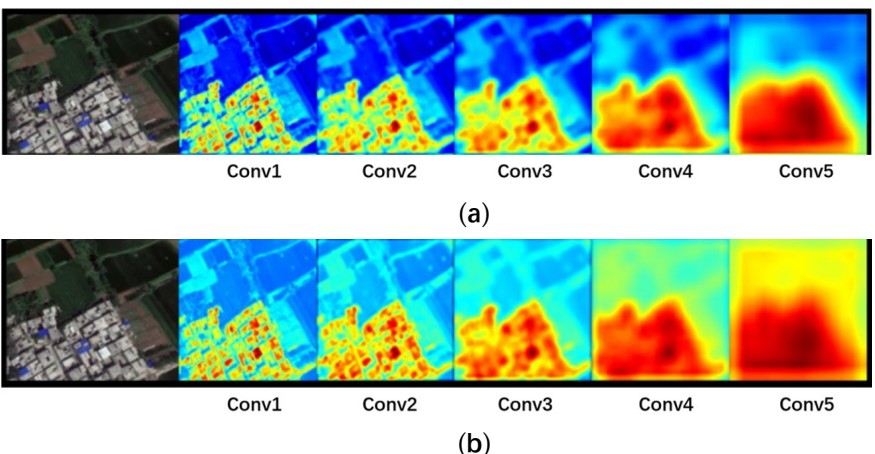

**Figure 2.** (**a**) is deformable convolution and (**b**) is traditional convolution feature maps of $3 \times 3$ convolution.

Next, a cross-attention [32] module was introduced between the third and fourth layers (*Pool3* and *Pool4*) to effectively capture local features and their global correspondence, enhancing feature representation. Afterward, the feature, which had undergone feature enhancement, was passed to the fourth deformable convolution layer. In this deformable convolution layer, to prevent loss of feature map information, the input feature map was iteratively sampled using average pooling. Additionally, the size of the feature map was adjusted to 1/4 of the input image to improve localization accuracy. This approach enlarged the receptive field and accurately localized targets. Finally, the output map of the last convolutional layer (*Conv4_3*) in the fourth layer was chosen as the feature map for selected key points. The calculation is shown in Formula (2), and the calculation flow is shown in Figure 3.

$$\mathbf{y}_i = \frac{\sum_{\forall j} f\left(\theta(\mathbf{c}_i)^T \phi\left(\mathbf{p}_j\right)\right) g\left(\mathbf{p}_j\right)}{\sum_{\forall j} f\left(\theta(\mathbf{c}_i)^T \phi\left(\mathbf{p}_j\right)\right)} \tag{2}$$

In the cross-attention computation, the input feature matrices *c* and **p**, where $C_i$ and $P_j$ represent the features at positions *i* and *j*, respectively. $\theta(\cdot)$, $\phi(\cdot)$, and $g(\cdot)$ represent linear embeddings, and $f(\cdot) = \exp(\cdot)$. In this calculation process, the function $f(\cdot)$ is used to measure the correlation between the features at positions *i* and *j*. The computed result $y_i$ represents a normalized summary of features through the Softmax [33] function, which

assigns weights to the features at all positions in $P$ based on their correlation with the cross-modal feature at position $i$. The matrix $Y$ composed of $y_i$ can integrate non-local information from **p** to every position in $C$. Finally, the output matrix $Z$ is obtained by adding matrices $Y$ and **p**, allowing for efficient backpropagation. In the output matrix $Z$, the feature at position $k$ summarizes the non-local correlation between the entire primary feature map and the position $k$ of the cross-modal feature map, as well as the original information from the primary feature map at position $k$.

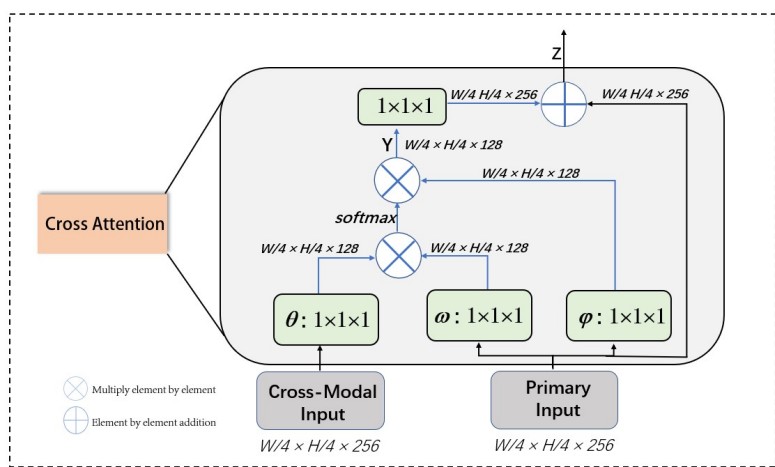

**Figure 3.** Cross-attention calculation process [32].

Then, we removed the fifth layer *(Conv5_1 + Conv5_2 + Conv5_3 + Pool5)* of VGG16 and the full connection layer at the end of VGG16. We considered that a large number of convolution operations in the convolution layer of the network would produce a certain number of negative outputs, and after the activation function of ReLU [34], these negative values would be replaced by 0, which would lead to the loss of a lot of feature information, and then would affect the feature detector. Therefore, in the network proposed in this paper, the ReLU activation function in all convolution layers was changed to Mish's [35] activation function.

In the aforementioned network design, if all the pixels in *Conv4_3* are used as features, it would lead to the issue of redundant features and high-dimensional feature space, which increases the computational cost and makes the model more complex and more prone to overfitting. To address these issues, feature selection is necessary. The channel-wise and spatial-wise maximum filtering strategy [20] is utilized to select the features with the best performance in the high-dimensional feature map while eliminating unnecessary features, thereby reducing feature dimensionality and retaining important features.

In order to make the model adapt to the images with large-scale variations, the concept of the discrete feature pyramid is introduced into the proposed model. Four discrete scale layers with 0.5, 0.75, 1.0, and 2.0 times resolution are adopted to adapt to the scale changes between the two images. The feature maps presented by each layer of the pyramid are cumulatively fused. Because the resolution of the pyramid is different, the low-resolution feature map needs to be linearly interpolated into the high-resolution feature map of the same size. In addition, by using the training data with certain scale differences, the model can adapt to the image with large-scale changes.

### 2.2. Feature Matching

In image matching, some physical constraints need to be observed. A feature point can only have at most one correspondence in another image; due to interference caused by light, shooting angle, etc., some feature points will not match. Therefore, an effective feature matcher should find corresponding feature points and be able to detect feature points that are not correctly matched. The query image $I_q$ and reference image $I_r$ are input into CMRM

to obtain their feature maps $F_q$ and $F_r$, executed key points are screened out to obtain $K_{pq}$ and $K_{pr}$, and the initial matching of $K_{pq}$ and $K_{pr}$ is completed through BFMatcher. After the initial matching, because of the multi-temporal remote sensing images and certain similar matching points in some scale spaces, it is inevitable that there will be mismatching. The RANSAC is a stable and reliable algorithm for eliminating wrong matching points. It estimates the mathematical model through continuous iteration. However, due to the large differences in multi-modal remote sensing images, the iterative method of RANSAC will be affected by this. In order to ensure that the algorithm is rational, this paper will carry out adaptive transformation on the traditional RANSAC algorithm. By designing an adaptive threshold method, it can reduce the impact on the matching results brought by artificially set thresholds. The specific steps for improvement are as follows:

- Step 1: Assuming that the feature point sets extracted from two images to be matched are $F_1$ and $F_2$, each feature point in $F_1$ is $D_{ij}$, the two points with the smallest Euclidean distance in $F_1$ are marked as $D'_{ij}$ and $D''_{ij}$, and the total number of feature points extracted from the images is $n$; then, the average set of their differences can be obtained.

$$G_D = \left( \frac{D'_{11} - D''_{11}}{2}, \frac{D'_{12} - D''_{12}}{2} \dots \dots \frac{D'_{nn} - D''_{nn}}{2} \right) \tag{3}$$

- Step 2: Select 10 points from the feature point set extracted in the previous step to establish a constraint equation and calculate its homography matrix.
- Step 3: Sort the difference average set $G_D$ extracted in step 1 Formula (3), in descending order, excluding the first 5% and the last 10% of data, and sum and average the remaining data as the constraint threshold of RANSAC. When the distance of each feature point to be matched is less than the constraint threshold, it is regarded as an interior point; otherwise, it is excluded.
- Step 4: When the number of interior points no longer changes, update the parameter model and end the iteration.

The adaptive threshold extracted by this method can adapt well to the differences between images, which effectively preserves high-quality matching points and enhances the stability of the RANSAC algorithm.

*2.3. Loss Function*

In the description of feature points, in order to achieve the effect of accurate matching, the feature character needs to be unique. Therefore, triplet loss is selected as the loss function in this paper. The triplet loss function punishes any unrelated descriptors that result in incorrect matches and enhances the uniqueness of the descriptors for more accurate results. Its calculation formula is shown in Formula (4):

$$\mathcal{L}(I_1, I_2) = \sum_{c \in \mathcal{C}} \frac{s_c^{(1)} s_c^{(2)}}{\sum_{q \in \mathcal{C}} s_q^{(1)} s_q^{(2)}} m(p(c), n(c)) \tag{4}$$

where $s_c^{(1)}$ and $s_c^{(2)}$ are the detection scores of a matching feature point in the reference image $I_1$ and the source image $I_2$. $p(c)$ and $n(c)$ represent positive distance and negative distance; $C$ is the set feature points in $I_1$ and $I_2$.

## 3. Experimental Results and Analysis

In order to verify the matching performance of the algorithm proposed in this paper, comparative experiments were conducted on the public dataset MRSI [36] and the self-collected dataset GeGZ, and ablation experiments were conducted on the HPathes dataset [37]. These datasets will be introduced in Section 3.1. The experimental design and evaluation indexes are introduced in Section 3.2. Then, in Section 3.3, experiments are introduced to compare a variety of (multi-temporal and multi-modal) remote sensing

matching algorithms, including deep learning-based CMM-Net [38], RIFT [39], DELF [40], and template matching-based CFOG [15] and CoFSM [36]. In Section 3.4, the registration performance of the proposed algorithm under different rotation angles was tested. The ablation experiment was verified in Section 3.5. Finally, an example of an intuitive result graph is given in Section 3.6.

### 3.1. Dataset

The dataset for comparison and evaluation consists of the following three parts:

- MRSI dataset: MRSI is a multi-modal remote sensing image database, which is composed of six mode pairs. A total of six groups of multi-temporal image experiments are set up, which are optic–optical (cross-time), infrared–optical, depth–optical, map–optical, sar–optical, and night and day. These images have different resolutions, ranging from $400 \times 400$ to $650 \times 650$. Each dataset contains 10 image pairs, producing a total of 60 MRSI pairs, with contrast loss and geometric deformation being the main variations between image pairs.
- GeGZ dataset: In order to enrich the diversity of the dataset, this paper collected a large number of remote sensing images by Google Earth Pro, ZY3 high-resolution optical satellite, and GF multi-spectral satellite. There are two main reasons for using the Google Earth dataset: (1) Google Earth can provide free historical images for many locations, from which we can select appropriate image pairs suitable for multi-temporal remote sensing registration. (2) Diversified Google Earth image datasets include atmospheric conditions and perspective changes. These images are representative of many variations in lighting conditions and seasonal conditions, which help train CMRM. In order to enrich the dataset, high-resolution images taken by different sensors, ZY3, GF1, and GF2, were also selected for training, which, to some extent, made up for the limitations of a single dataset. These datasets include rivers, coastlines, roads, farmland, forests, and urban and rural buildings. The dataset is divided into six groups, with each group containing 358 pairs of multi-temporal images. In the experiment, various data augmentation techniques, such as random cropping, random rotation, and color space enhancement, were applied to the dataset, expanding the dataset to a total of 6000 pairs. The training set and validation set have a ratio of 8:2 in terms of the number of samples. These images have sizes ranging from $583 \times 583$ to $1965 \times 1024$ pixels and resolutions ranging from 0.5 m/pixel to 2 m/pixel. Examples of image data from the above six groups are shown in Figure 4 and Table 1.
- HPatches: In order to further analyze the performance of each module of the algorithm presented in this paper, this paper conducts experimental verification on the public dataset HPatches (Homography Patches), which consist of two sets of images, one consisting of 59 pairs of viewpoint transformations and the other of 57 pairs of illumination variations. The homography between the reference image and the target image was carefully calibrated. It has been cited in many registration tasks due to its diversity, real data origin, large scale, and multi-tasking characteristics [41,42].

**Table 1.** A specific description of each pair of images in Figure 4.

| Image Pair | Reference Image | | Source Image | | Image Characteristics |
|---|---|---|---|---|---|
| | Image Source | Time | Image Source | Time | |
| Pair 1 (a) (b) | ZY3(MUX) | 2020 | ZY3(NAD) | 2022 | The image pair has a high resolution, and the feature changes greatly. |
| Pair 2 (c) (d) | Google Earth | 2017 | Google Earth | 2022 | The illumination difference between ground objects is obvious. |
| Pair 3 (e) (f) | Google Earth | 2018 | Google Earth | 2022 | The images are selected from urban areas, and the background obviously changes. |

**Table 1.** *Cont.*

| Image Pair | Reference Image | | Source Image | | Image Characteristics |
|---|---|---|---|---|---|
| | Image Source | Time | Image Source | Time | |
| Pair 4 (g) (h) | Google Earth | 2016 | Google Earth | 2020 | Image time span is large; there is a certain change in perspective. |
| Pair 5 (i) (j) | GF1(WFV2) | 2018 | GF2(PMS2) | 2021 | Image detail texture is different; structure is similar. |
| Pair 6 (k) (l) | GF2(PMS1) | 2016 | GF2(PMS2) | 2021 | The image is affected by fog and other weather conditions, and there is noise. |

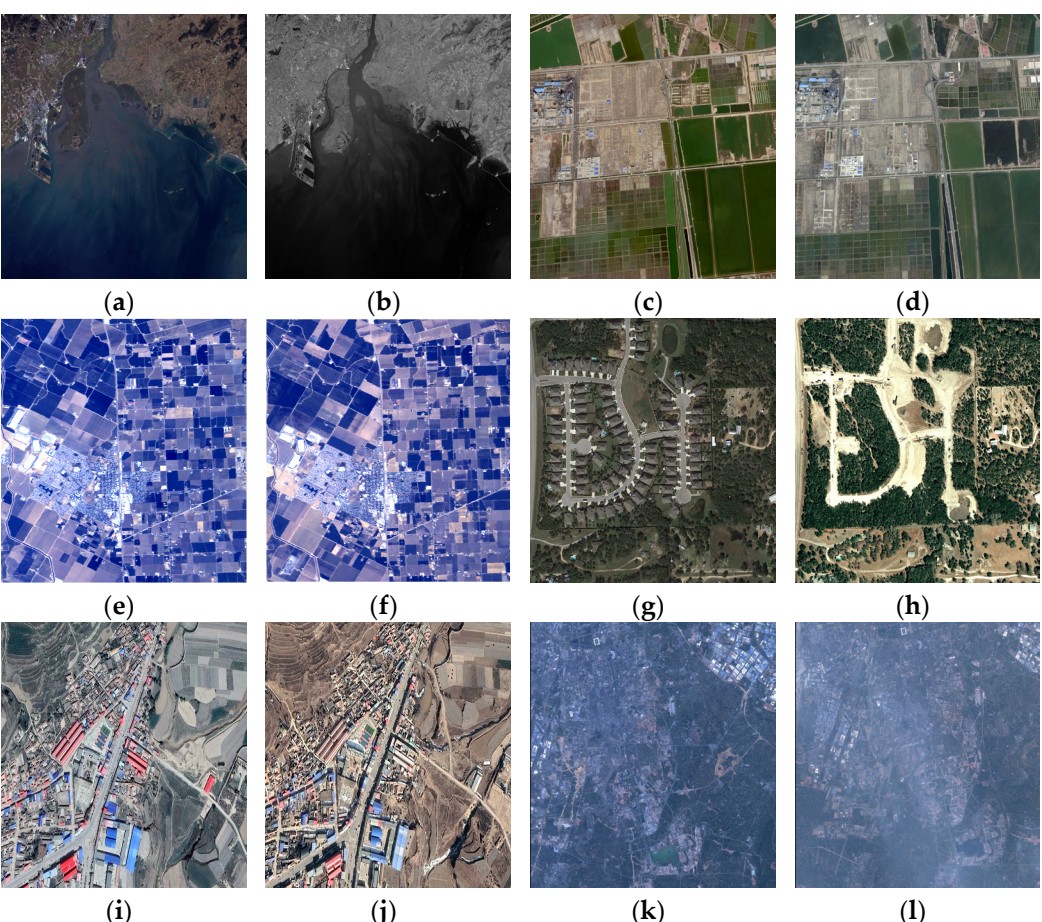

**Figure 4.** (**a**–**l**) are six groups of image pairs selected from GeGZ datasets.

### 3.2. Experimental Setting

3.2.1. Training

The experimental platform and system settings in this paper are as follows: the CPU is an Intel (R) Core (TM) i7-7500U2, 70GHz, and the GPU is NVIDIA GeForce RTX 3090; the operating system is Ubuntu 20.04, pytorch1.8 deep learning framework is used, and the Python version is 3.7. The gradient optimization algorithm uses stochastic gradient descent, SGD), and the initial learning rate is 0.001. The training process of each experimental network model is the same. The model training is performed in batches of eight sample pairs, with a maximum of 50 epochs. Training is stopped when the difference in the loss function is less than 0.01. The maximum number of rounds of model training is 50 rounds, and the training is stopped when the difference of loss function is less than 0.01. The training process of the model is shown in Algorithm 1:

---

**Algorithm 1:** Training procedure.

---

*Input*: Training multi-temporal dataset $D$
Initialize model $M_w$ parameters using pre training weights
*Output*: Trained model $M_w$
*for* epochs *do*
   *for* $(I_p,\ I_c)$ *in D* *do*
       # Construct two inputs
       $I_p =\ preprocess(I_p)$
       $I_c =\ preprocess(I_c)$
       # Forward propagation
       $\check{I}_p =\ DeVgg(I_p)\ +\ cross\text{-}att(I_p,\ I_c)$
       $\check{I}_c =\ DeVgg(I_c)\ +\ cross\text{-}att(I_c,\ I_p)$
       # Compute loss
       $L\ =\ L\left(\check{I}_p,\ \check{I}_c,\ I^{gt}\right)$
       # Backward propagation and Update parameters
       $\omega = \omega - lr * \frac{dl}{dw}$
   *end*
   # Validate the model
*end*

---

### 3.2.2. Evaluation

In this paper, we use the number of correct matches (NCM), which has been widely applied in other matching tasks [43,44], as well as the number of feature points, root mean square error (RMSE) of matched points, matching time, and mean matching accuracy (MMA) as evaluation metrics. Their descriptions are as follows:

Formula (5), judging the number of correct matching points (NCM), is as follows:

$$\mathrm{Corr}\,(\boldsymbol{n}) : \|\,n_1 - n_2\,\| \le \varepsilon \tag{5}$$

The position of a feature point in reference image A is $n_1$, and the position of a feature point in the source image B is $n_2$. The homography matrix can map the coordinates in A into B. When the distance between the mapped $n_1$ and $n_2$ is less than the precision threshold, it is considered a correct match. This index can reflect the performance of the registration algorithm.

The root mean square error (RMSE) of a matching point is calculated as follows in Formula (6):

$$\mathrm{RMSE} = \frac{1}{\mathrm{NCM}} \sum_i \|\,\boldsymbol{H}(n_1) - n_2\,\| \tag{6}$$

The RMSE of the matching point indicates the degree of deviation from the true value. The smaller the RMSE, the higher the measurement accuracy. H indicates the transformation model of the two images after homography matrix calculation.

In the HPatches dataset, MMA with an error threshold of 3–5 is taken as the evaluation index, and MMA is the average value of correct matching in image pairs considering multiple pixel error thresholds.

### 3.3. Comparative Experiments

The five selected methods are excellent image-matching algorithms based on local features. RIFT and CFOG have been the representative multi-modal registration algorithms in the past two years. DELF and CMM-Net are based on convolutional neural networks, and perform well in the registration tasks with a chaotic background and partial occlusion. The comparison results are shown in Tables 2 and 3.

**Table 2.** Average NCM in the MRSI dataset.

| Method | Pair 1 | Pair 2 | Pair 3 | Pair 4 | Pair 5 | Pair 6 |
|---|---|---|---|---|---|---|
| RIFT [39] | 506.6 | 515.5 | 425 | 386.2 | 266.3 | 375.5 |
| CoFSM [36] | 595.9 | **641.3** | **566.9** | 400 | 351.6 | **435.4** |
| CMRM | **666.2** [1] | 629.6 | 413.8 | **462.7** | **384** | 369.7 |

[1] Number in bold represents the optimal value of NCM.

**Table 3.** Comparison of experimental results in the GeGZ dataset.

| Matching Method | Evaluation Index | Pair 1 | Pair 2 | Pair 3 | Pair 4 | Pair 5 | Pair 6 |
|---|---|---|---|---|---|---|---|
| RIFT [39] | NCM | **944** [1] | 232 | **1205** | 22 | 790 | 904 |
| | Feature Points | 2361 | 1321 | 2091 | 1618 | 2229 | 1787 |
| | Times/s | 23.88 | 15.62 | 31.94 | 32.29 | 21.17 | 18.47 |
| CFOG [15] | NCM | 127 | 74 | 105 | 26 | 84 | 4 |
| | Feature Points | 163 | 284 | 495 | 199 | 192 | 245 |
| | Times/s | 5.38 | 6.37 | 5.92 | 7.77 | 6.38 | 5.66 |
| CMM-Net [38] | NCM | 721 | 256 | 1051 | 15 | 922 | 896 |
| | Feature Points | 1423 | 986 | 2195 | 199 | 1182 | 1985 |
| | Times/s | 3.90 | 2.81 | 3.83 | 5.17 | 2.88 | 4.62 |
| DELF [40] | NCM | 675 | 152 | 42 | 9 | 910 | 981 |
| | Feature Points | 2111 | 571 | 2310 | 1721 | 2102 | 1523 |
| | Times/s | 14.63 | 7.71 | 18.26 | 7.81 | 10.60 | 9.61 |
| CMRM * | NCM | 782 | **1549** | 1095 | **52** | **965** | **1252** |
| | Feature Points | 2135 | 2699 | 2307 | 2059 | 1823 | 2462 |
| | Times/s | 2.48 | 1.43 | 1.22 | 1.41 | 2.91 | 3.01 |
| CMRM | NCM | 749 | 1420 | 508 | 36 | 890 | 1204 |
| | Feature Points | 2135 | 2699 | 2307 | 2059 | 1823 | 2462 |
| | Times/s | 2.44 | 1.52 | 1.26 | 2.84 | 2.23 | 2.88 |

* Represents the RANSAC adaptive threshold. [1] Number in bold represents the optimal value of NCM.

### 3.3.1. Comparative Experiments on MRSI

According to the analysis of Table 2 and Figure 5, the mean NCM of all image pairs in RIFT is 412.52, and the mean standard deviation is 5.294. RIFT has high matching accuracy and robustness, but high computational complexity requires a lot of computing resources and time. The average NCM of all image pairs in CoFSM is 498.52, and the average standard deviation is 2.956. The matching effect in these six groups of experimental data is excellent. This is because CoFSM has high registration accuracy and computational efficiency by transforming the image registration problem into a combinatorial optimization problem, but it is more sensitive to input image illumination and noise interference. The average NCM of all image pairs of CMRM is 486.83, and the mean standard deviation is 3.118. CMRM performs well in multi-temporal images and map-optical images, but in depth-optical and sar-optical datasets, the average RMSE of CMRM is 0.188 higher than CoFSM, and the registration effect is slightly lower.

### 3.3.2. Comparative Experiments on GeGZ

Through the analysis of Table 3 and Figure 6, it can be seen that in the six groups of experiments, the CMRM algorithm can all match matching points with certain correct values, and the feature points extracted by CMRM are evenly distributed and accurately positioned. The average RMSE value of CMRM is lower than that of the other four registration methods, which indicates that the registration effect in this experiment is good and the information utilization rate between images is high. Both RIFT and DELF can obtain more matching points, but the matching of RIFT is significantly reduced in the image with larger scale transformation. CMM-Net also works well in images with changing

backgrounds, and CFOG makes only a small number of matches. The experimental results are shown in Figure 7 as follows. It shows that among the above six matching algorithms, the CMRM algorithm has the best performance in matching.

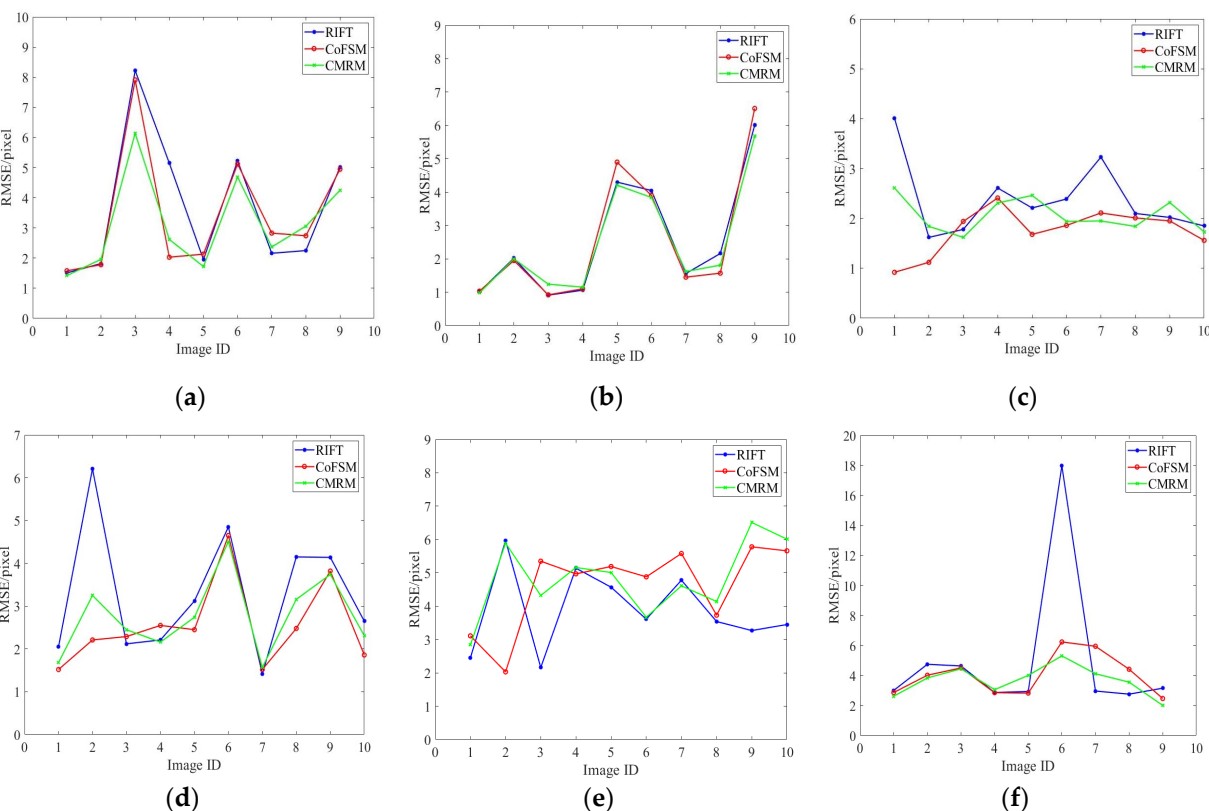

**Figure 5.** (**a**–**f**) are the RMSE metric results of six sets of MRSI datasets.

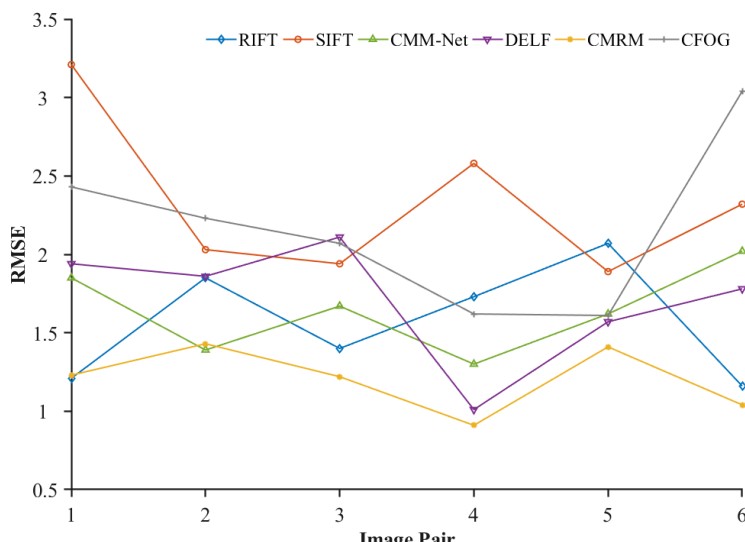

**Figure 6.** Comparisons on RMSE metric in GeGZ dataset.

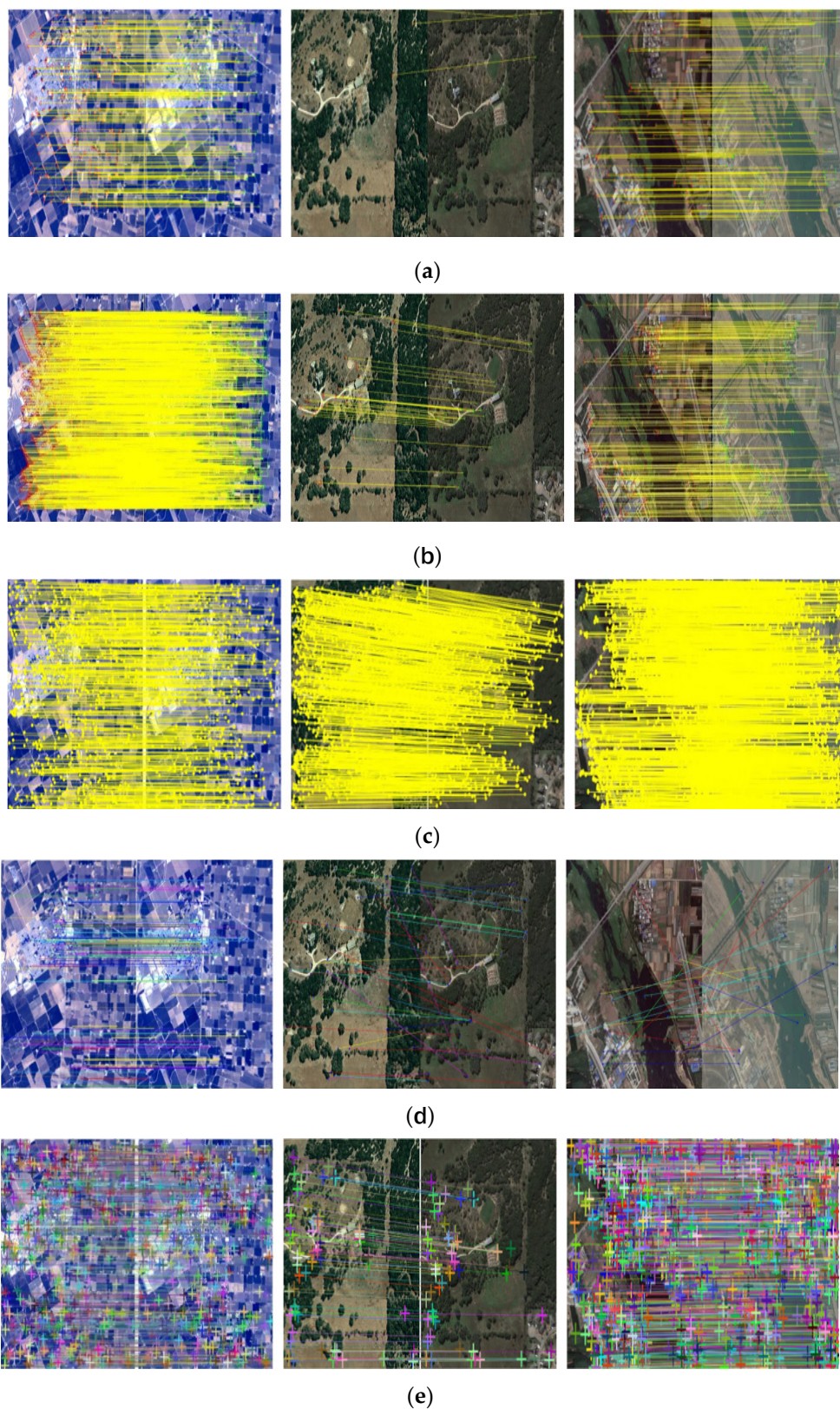

**Figure 7.** (**a**–**e**) are the comparisons of matching results of CFOG, RIFT, CMRM, DELF, and CMM-Net in three groups of image pairs.

*3.4. Rotation Adaptation Experiment*

To verify the ability of CMRM to align images with rotational changes, four sets of images are randomly selected from the GeGZ dataset, and then each set of images is rotated

from 20° to 80° at 20° intervals to generate four rotated images. Finally, 16 new matching result maps are obtained and used to evaluate the performance of the proposed algorithm in terms of rotation adaptation. Meanwhile, the image to be aligned is rotated by 80° in 1° steps, and the cosine distance between the features extracted from the rotated image and the features on the reference image is calculated as the similarity index. The results of the experiment are shown in Figure 8a. The NCM values of the four sets of rotated images made are shown in Figure 8b. One set of experimental matching results is shown in Figure 9.

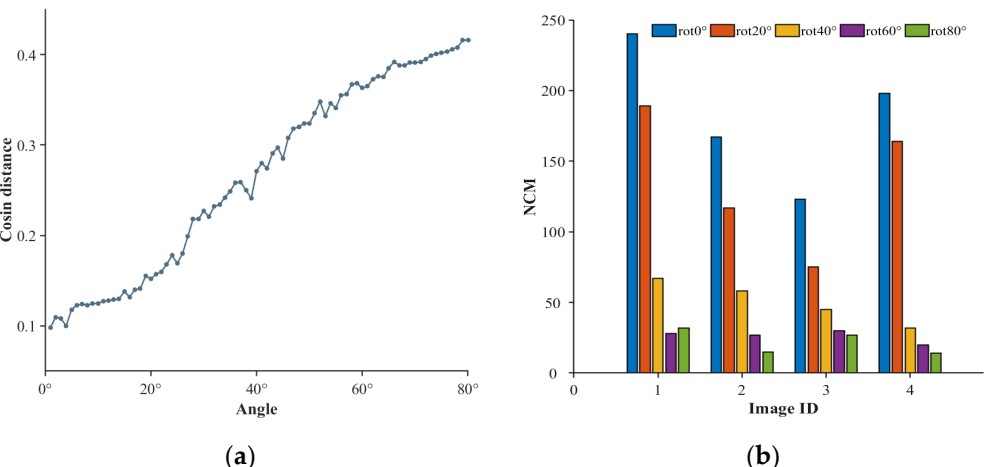

(a)            (b)

**Figure 8.** (**a**) is cosine distance and (**b**) is rotation invariance experiments.

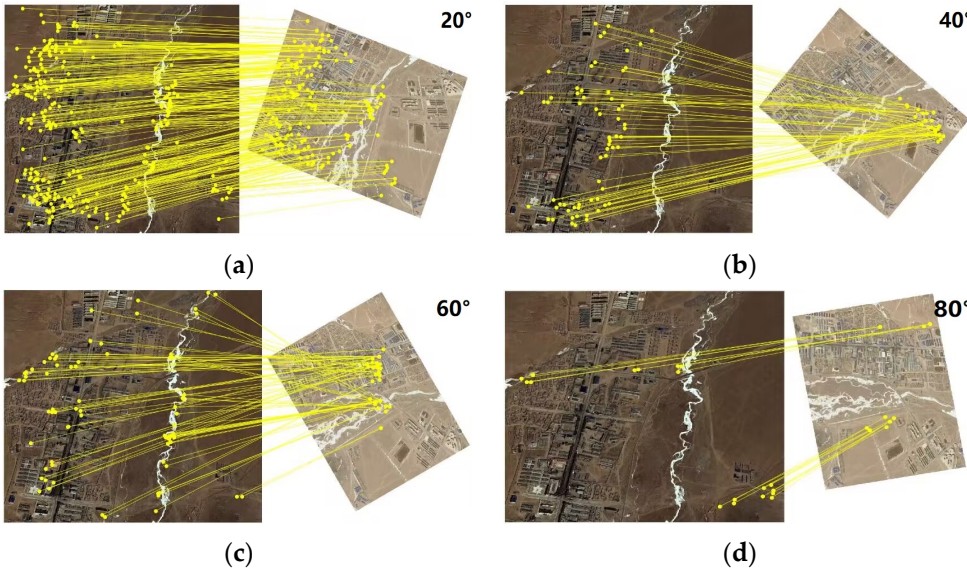

**Figure 9.** (**a**–**d**) are the matching results for rotations of 20°, 40°, 60°, and 80°.

By analyzing the above results, it can be observed that overall, the algorithm maintained a good level of feature similarity within a certain range of angles (30°). This is because the network learned the invariance to small rotations during the training process. However, when the rotation angle exceeded 30°, the NCM experienced an average decrease of over 50%, and the cosine distance between features increased significantly.

### 3.5. Ablation Experiment

3.5.1. Functional Module Ablation Experiment

In order to further analyze the function of each module in this algorithm, ablation experiments with a threshold of 3–5 are carried out in the HPatches dataset, and the results are shown in Table 4; a set of experimental results are shown in Figure 10.

**Table 4.** Ablation results in the HPatches dataset.

| Method | Cross-Attention | DeVgg | MMA |
|---|---|---|---|
| | | | ($\varepsilon$=3px | 4px | 5px) |
| CMRM | $\checkmark$ | $\checkmark$ | **0.672** [1]**/0.710/0.785** |
| | $\checkmark$ | | 0.661/0.706/0.721 |
| | | $\checkmark$ | 0.657/0.673/0.749 |
| Baseline | | | 0.421/0.456/0.571 |

[1] Numbers in bold represent the optimal values of MMA.

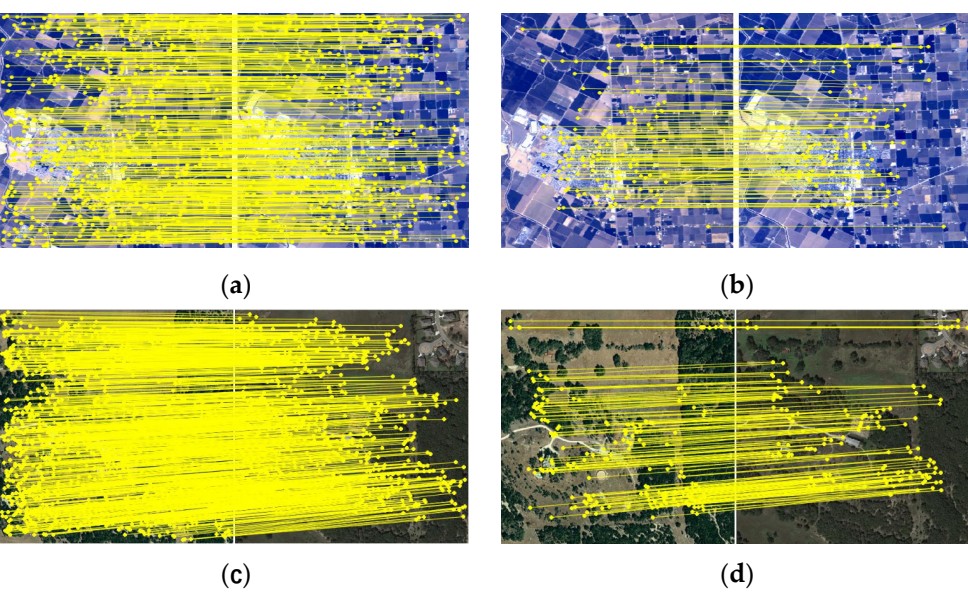

**Figure 10.** (**a**–**d**) are a comparison of the results of adding Cross Attention and DeVgg and not adding Cross Attention and DeVgg.

(1) When only cross-attention is added, the MMA of the dataset is increased by 0.24, 0.248, and 0.147, respectively, compared with that of the baseline network, which effectively verifies that the network's ability to express the spatial correspondence between the features of multi-temporal remote sensing images is improved when the attention module is added; (2) with the addition of the DeVgg module, the ability of feature extraction and registration is obviously higher than that without the DeVgg module, which verifies the effectiveness of DeVgg. The main reason is that during the training process, the training data have obvious background changes, which makes the features extracted by the network have a robust expression, and strengthens the ability of feature expression of deformation. (3) By combining cross-attention and DeVgg, the MMA value of the public dataset is increased by 0.251, 0.254, and 0.214, respectively, compared with that of the baseline network. As can be seen from Figure 10, when the deformable convolution layer and attention surrender were added, the registration result of the network was significantly improved.

3.5.2. Experiments on Different Backbone Networks

CMRM selected the VGG16 model pre-trained on the ImageNet dataset as the backbone network. In order to verify the accuracy of the selection, different backbone feature extraction networks were evaluated on the HPatches dataset, ResNet50 and MobileNet

V2 pre-trained on the ImageNet dataset were tested, and the dataset was added with deformable convolution. In the original VGG16, the resolution of the output feature map was 1/8, but in this paper, in order to improve the positioning accuracy, the resolution was expanded from 1/8 to 1/4. Similarly, the original output of ResNet was expanded from 1/16 to 1/4. The results are shown in Figure 11.

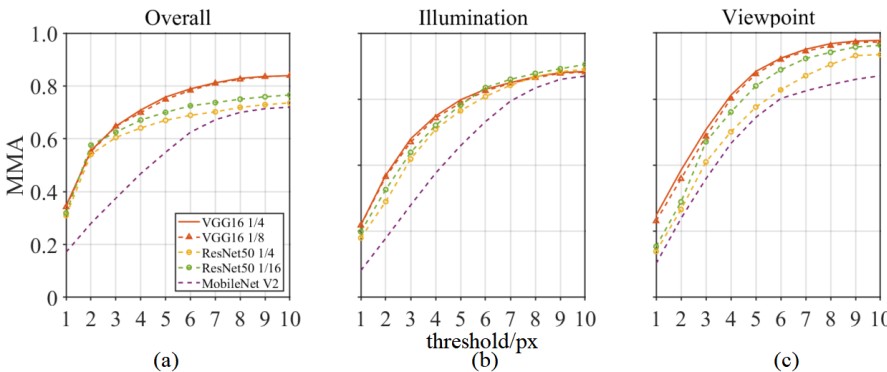

**Figure 11.** (**a**–**c**) show the matching MMA values and matching point values of different backbone models on the HPatches dataset, demonstrating the uneven matching accuracy achieved under different thresholds.

Overall, different backbone networks exhibit significant variations in MMA on the HPatches dataset. MobileNet V2 performs the worst under different threshold conditions, while VGG16 achieves the best results at 1/8 and 1/4 resolutions. Considering the impact of lighting conditions, we conducted experiments with different threshold conditions, and the results show that VGG16 outperforms MobileNet V2 at 1/8 and 1/4 resolutions, and it also outperforms ResNet when the threshold is less than 6. Considering the effect of viewpoint changes, the experimental results with different threshold conditions are shown in Figure 10c, indicating that VGG16 outperforms ResNet and MobileNet V2 at 1/8 and 1/4 resolutions.

### 3.6. Examples of Matching Results

Figure 12 further demonstrates the feature extraction capability of CMRM. The selected six pairs of images are from the GeGZ dataset, and their specific descriptions are shown in Table 1. The image pairs in Figure 12a are from different sensors, and the images show irregular changes in water systems. The zoom-in results show that CMRM can still achieve good matching in areas with significant changes. The image pairs in Figure 12b–d are all from Google Earth. The content of the image pair in Figure 12b is similar in structure and distributed uniformly in farmland, with obvious perspective differences. Although only a small number of feature points are extracted, a high degree of matching can still be achieved. The image pairs in Figure 12c contain a large amount of arable land. Under highly similar shapes, CMRM can still find many corresponding points. The image pair in Figure 12d has a large time span and significant background differences. CMRM can only find a small number of corresponding points; however, compared to other algorithms, it still achieves good results. The image pairs in Figure 12e are taken from urban areas and contain many urban buildings and roads with many detailed texture differences. CMRM shows good matching performance. The image pairs in Figure 12f have noise due to weather conditions, but CMRM can still find a large number of feature points. From Figure 11, it can be seen that the feature points extracted by CMRM are distributed evenly and accurately locate the corresponding areas in the images, indicating that CMRM has good feature extraction capabilities.

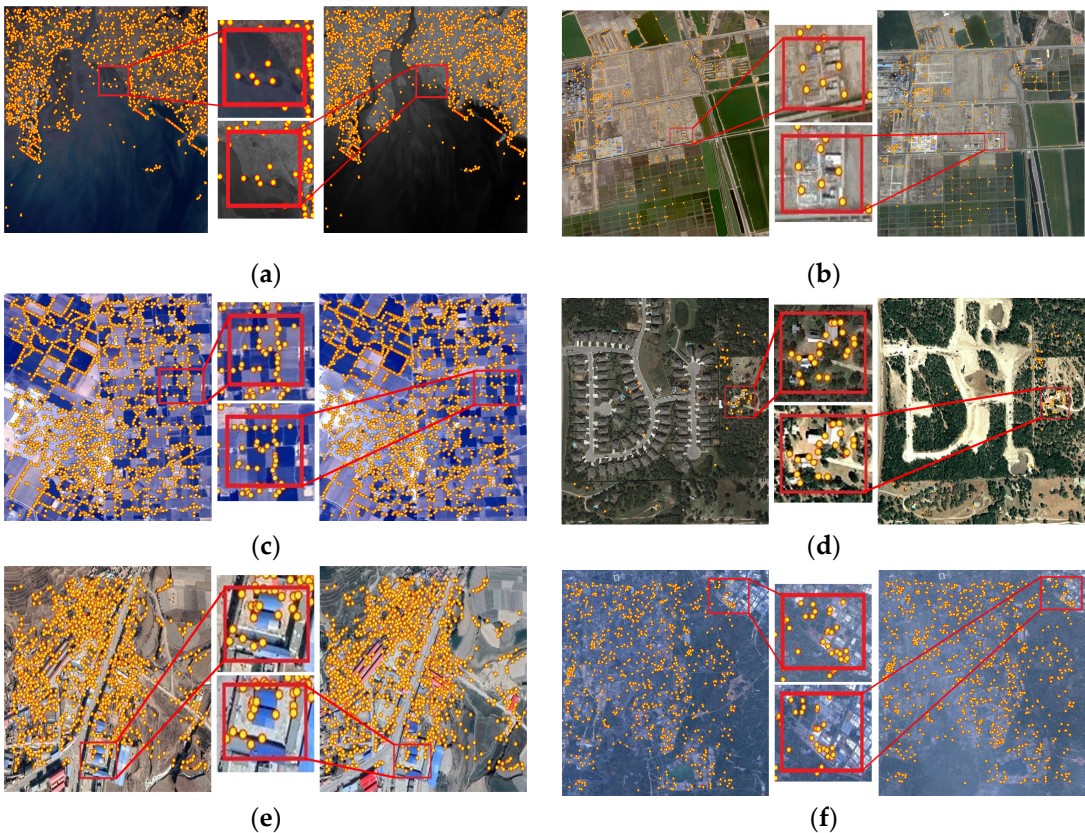

**Figure 12.** (**a**–**f**) are the feature point detection results of the six groups of experiments and the corresponding enlarged part, respectively.

## 4. Conclusions

The present study proposed a convolutional neural matching network based on cross-attention and deformable convolutions. By leveraging the cross-attention, it enhanced the spatial correspondence between local and global features. Additionally, the network utilized deformable convolution layers to improve the representation capability of deformation features. We also introduced an adaptive threshold constraint in the feature removal stage to obtain high-quality matching points.

Experimental results demonstrated that the proposed algorithm was suitable for multi-modal remote sensing image registration tasks with significant background variations and certain viewpoint changes. Furthermore, compared to other multi-modal matching algorithms, it exhibited advantages in terms of feature point extraction quantity, matching accuracy, and time. However, during the training process of the proposed network, a precise localization model was required to eliminate scale and geometric differences between images, which increased the training time. In future research, the matching performance of this algorithm will be further investigated under large rotation angles and different modal transformations.

**Author Contributions:** P.C. designed the network architecture, analyzed data, wrote the paper, and experimentally validated technical routes; Y.F. designed technical routes, designed experiments, provided funding support, and made major revisions to academic content; J.H. analyzed data, provided funding support, and made revisions to academic content; B.H. selected topics, determined the paper's framework, analyzed and interpreted data, and made major revisions to academic content; X.W. and J.Z. made major revisions to academic content. All authors have read and agreed to the published version of the manuscript.

**Funding:** This research was funded by the Sichuan Natural Science Foundation (No. 2022NS-FSC0907, No. 2023NSFSC0243), Sichuan Science and Technology Program (No. 2023YFQ0072, No. 22QYCX0082), Science and Technology Major Project of Sichuan (No. 2019DZX0005), and Project of Innovation Ability Enhancement of Chengdu University of Information Technology (No. KYQN202219).

**Data Availability Statement:** The code used in this study can be obtained by contacting the first author. The data sources of the GeGZ are https://www.earthol.com/ (accessed on 19 September 2022). and http://www.sasclouds.com/chinese/normal/; The data source of the MRSI is https://ieee-dataport.org/documents/cofsm (accessed on 14 March 2022).

**Acknowledgments:** We sincerely thank the authors of RIFT, CFOG, DELF, CoFSM, and CMM-Net for providing their codes to facilitate the comparative experiments, and thank the anonymous reviewers for their valuable suggestions in improving the quality of this paper.

**Conflicts of Interest:** The authors declare no conflict of interest.

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
