# Peer review of "An Adaptive Remote Sensing Image-Matching Network Based on Cross Attention and Deformable Convolution"

_electronics, doi:10.3390/electronics12132889_

Round 1

Reviewer 1 Report

The reasoning for employing VGG16, a CNN architecture, is unclear given the availability of more performant alternatives.

Additional experiments could be conducted to explore the performance of different base models and their compatibility with the proposed approach.

The problem is comprehensively and effectively elucidated.

The bibliography is appropriately and accurately utilized.

The article exhibits a well-organized structure and presents information in a clear and accessible manner, facilitating ease of reading.

Reviewer 2 Report

In his paper, authors presents an improved deformable CNN network for image registration applied to satellite remote sensing. The algorithm is mainly based on a VGG16 CNN architecture that has been modified introducing two deformable convolution layers to accommodate spatial deformations. The feature point descriptor is obtained from the output of the fourth level convolutional layer (conv4-3). This approach bears similarities to the existing D2Net network, but the authors introduce a cross-attention mechanism that incorporates feedback from the correspondence stage.

Section 2.1 lacks some clarity in explaining the feature selection process. It is based on the VGG16 conv4_3 output layer, which has dimensions of 28x28x512 (similar to D2-Net). Consequently, it produces a 512-feature descriptor for a dense spatial matrix of 28x28. The authors propose a feature selection technique that employs maxima filtering both spatially and channel-wise. In D2-Net, an additional phase involving the SIFT detector is used to refine the localization of feature points. However, the paper does not provide detailed information on whether they utilize this low-resolution localization estimation (28x28) for feature estimation and how it impacts the estimation of registration transformation based on correspondences.

The correspondence phase relies on the Brute Force algorithm with outlier suppression using a standard RANSAC algorithm.

The background and introduction sections are  well-written, the proposed method is fairly well described  and the research methodology adequate. 

The paper introduces some novel approaches to enhance algorithm performance, such as the introduction of cross-attention feedback between layers 3 and 4 to establish correspondences between registered images.

The results section presents some significant performance improvements compared to existing algorithms using various public databases. The proposed algorithm can be considered an interesting contribution as an improved method

Format of the paper and English grammar are adequate. The manuscript is well organized and redacted. The mathematical background and references are adequate. Bibliography is extensive and up-to-date.

There are some minor grammar and vocabulary errors in the paper that need to be corrected (see comments below).

Lines 10-13: The first paragraph is poorly written and lacks clarity. Please rephrase it to convey the intended meaning.

Line 19: Replace the term "brought" with a different past participle, such as "produced," in the sentence "background changes brought by time."

Line 69: Replace the term "artificial" with "manual" in the sentence "require artificial design of feature descriptors."

Line 83: Rephrase the sentence “. It directly optimized features …”.  For example:  “this proposal directly optimize features…”

Line 96: Completely rephrase the sentence "improving the robustness of the algorithm to outliers and other sources of noise in the data" as the term "to outliers" is grammatically incorrect in this context.  

Reviewer 3 Report

The authors propose a “An Adaptive Remote Sensing Image Matching Network Based 2 on Cross Attention and Deformable Convolution”, which attempts to address the problem to extract features to effectively matching accuracy of multi-modal remote sensing images.

The authors correctly introduce the problem and give a good literature review of today’s algorithms applied to the topic.

After reading the paper I would like the authors comments related to:

-The comparison of traditional algorithms and their  opposite challenging deep learning algorithms leads to transferring the complexity to datasets. From the paper itself, the number of image pairs is sufficient.

The datasets used to train, test and validation seems to have not enough images. It was implemented any data augmentation strategy.

-How the input convolution layer in (vvg16) adapt to optimize the features extraction with to very different image illumination conditions?

-Why the authors don’t use other deep leaning algorithms, data augmentation technics and well balanced dataset to extensively train a DL model?

-The present research work is a very interesting in the computer vision field, but from the paper itself I can’t conclude what the authors conclude, i.e., the experimental section must have a more complete approach.

As finally, can your proposed method be generalized to different illuminations and textures conditions with no lack of performance?

As a conclusion of the review, the authors are invited to address my questions preferably. The paper is well written and for this reason my recommendation is that it may be accepted.

None
